# Scale-Free Distribution of Oxygen Interstitial Wires in Optimum-Doped HgBa$_2$CuO$_{4+y}$

**Gaetano Campi** [1,*] , **Maria Vittoria Mazziotti** [2,3], **Thomas Jarlborg** [2,4] **and Antonio Bianconi** [1,2,*]

1    Institute of Crystallography, CNR, Via Salaria Km 29.300, 00015 Monterotondo, Italy
2    Rome International Center for Materials Science Superstripes RICMASS, Via dei Sabelli 119A, 00185 Rome, Italy
3    Department of Mathematics and Physics, University Roma Tre, Via della Vasca Navale 84, 00146 Rome, Italy
4    DQMP, University of Geneva, 24 Quai Ernest-Ansermet, CH-1211 Geneva, Switzerland
*    Correspondence: gaetano.campi@ic.cnr.it (G.C.); antonio.bianconi@ricmass.eu (A.B.)

**Abstract:** Novel nanoscale probes are opening new venues for understanding unconventional electronic and magnetic functionalities driven by multiscale lattice complexity in doped high-temperature superconducting perovskites. In this work, we focus on the multiscale texture at supramolecular level of oxygen interstitial (O-i) atomic stripes in HgBa$_2$CuO$_{4+y}$ at optimal doping for the highest superconducting critical temperature ($T_C$) of 94 K. We report compelling evidence for the nematic phase of oxygen interstitial O-i atomic wires with fractal-like spatial distribution over multiple scales using scanning micro- and nano-X-ray diffraction. The scale-free distribution of O-i atomic wires at optimum doping extending from the micron down to the nanoscale has been associated with the intricate filamentary network of hole-rich metallic wires in the CuO$_2$ plane. The observed critical opalescence provides evidence for the proximity to a critical point that controls the emergence of high-temperature superconductivity at optimum doping.

**Keywords:** oxygen interstitials; quantum wires; critical opalescence; high-temperature superconductivity; scanning X-ray diffraction; lattice effects

## 1. Introduction

The true nature and the key role of the complexity at nanoscale in high-temperature superconductors [1–8] and related systems [9] have sparked growing interest in the condensed matter research field, since the discovery of unconventional overdoped perovskites [10,11]. The development of novel synchrotron radiation sources and imaging techniques on the basis of focusing on the X-ray beam down to the nanoscale allow for the visualization of multiscale inhomogeneities of the supramolecular structure [12–15]. In this way, scale invariant textures have been observed in lattice, spin, and charge degrees of freedom in strongly correlated oxide perovskites [16–21] and at metal-insulator transitions [22]. These results have shown that scale invariance [23,24] in granular matter near a critical point has been favoring quantum coherence in high-temperature superconductivity [25–29]. Lattice complexity beyond the average crystalline structure effects in the mechanism of high $T_c$ superconductivity at optimum doping is attracting high interest. In particular, the role of oxygen interstitials and vacancies spatial disposition has a strong impact on the electronic properties of superconducting perovskite materials [30–32]. The texture of oxygen interstitials (O-i) related with their self-organization due to their large mobility in the space layers above about 200 K was detected by advanced scanning micro-X-ray diffraction, in layered perovskites, such as nichelates La$_2$NiO$_{4+y}$ [33,34] and in several families of cuprate high-temperature superconductors (HTS), such as doped La$_2$CuO$_{4+y}$ (La214) [35–42], YBa$_2$Cu$_3$O$_{6+y}$ (Y123) [43–45], and Bi$_2$Sr$_2$CaCu$_2$O$_{8+y}$ (Bi2212) [46–49]. In this work, we focus on HgBa$_2$CuO$_{4+y}$ (Hg1201) at optimum doping [50–60], which is the simplest member of the mercury-based cuprate compounds that provides the highest

superconducting transition temperature. $HgBa_2CuO_{4+y}$ has a simple tetragonal average structure with average Cu-O bond length of 194 pm [54], showing self-organization of a nematic phase with co-existing O-i atomic stripes [56,57] running in the a-direction (100) and b-direction (010) of the ab plane. The presence of anisotropic structures at room temperature, and the understanding of their properties, are crucial to explain the still elusive room temperature nematicity observed in HTS [61,62]. However, while the nanoscale phase separation made of first puddles rich in O-i stripes anticorrelated with second puddles that show short-range charge density waves (CDW) has been visualized by scanning micro-X-ray diffraction [57], the imaging of the nanoscale phase separation on the mesoscale (100–1000 nm) in Hg1201 is still missing. Information on this intermediate scale appears to be quite important, since the oxygen interstitials form domains with a dimension ranging from few, up to tens of nanometers. Therefore, higher resolution in real space is required for the X-ray diffraction measurements to image oxygen interstitials ordering on the mesoscale. Thanks to the advances in X-ray, focusing on optics is nowadays possible to cover this length scale connecting the microscopic to the nano- and atomic-world using beams down to 100 nm in scanning mode.

In this work, we have used scanning micro-X-ray diffraction (SµXRD) with a focused beam of $1 \times 1$ µm$^2$ joint with scanning nano-X-ray diffraction (SnXRD) with a focused beam of $100 \times 300$ nm$^2$ for directly imaging the texture formed by the oxygen interstitials in Hg1201 at microscale and mesoscale. We exploited spatial statistic tools, such as the probability density function of O-i populations, spatial correlations, and connectivity to describe quantitatively the O-i textures from micron to mesoscale. The results confirm the power law behavior of the oxygen interstitials organized in stripes at micrometric scale in Hg1201 [57], which was observed first in $La_2CuO_{4+y}$ at optimum doping [35,39]. In addition, at improved resolution, we have been able to visualize a percolation interface between the insulating CDW puddles and the oxygen wires [61,62].

## 2. Results

The $HgBa_2CuO_{4+y}$ single crystals have been grown with a final oxygen treatment to establish the y concentration of oxygen interstitials of about y = 0.12 showing the superconducting optimum critical temperature ($T_C$) of 94 K [51–53]. The crystal structure has been determined by standard X-ray diffraction. The structure, schematized in Figure 1a, has a tetragonal P4/mmm space group symmetry, and the unit cell parameters are: a = b = 3.874 Å, c = 9.504 Å at room temperature in agreement with reference [54].

Nematic phase of atomic oxygen wires has been investigated by high-energy X-ray diffraction measurements performed at the BW5 beamline of the DESY Synchrotron in Hamburg. The results provide compelling experimental evidence for the location of the oxygen interstitial (O-i) content, y, which occupies the interstitial site $\frac{1}{2}$, $\frac{1}{2}$, and 0 in the basal plane of $HgBa_2CuO_{4+y}$. The arrangement of O-i gives rise to intriguing diffraction features constituted by clear diffuse streaks crossing the Bragg peaks along all the three crystallographic directions [56–58]. In Figure 1b, we show a typical diffuse streak between the (3,3) and (3,4) reflections in the hk diffraction plane. Figure 1c shows the streak profile in red and the background in black. The normalized X-ray diffraction profile is shown in the lower panel, where the diffuse streak intensity has been quantified by the difference between the Bragg peaks profile and the background. In this way, we obtained the streak intensity, due to the O-i stripes. The connection of the observed streaks with the O-i arrangement in stripes has been proved by calculating the X-ray diffraction pattern of a $HgBa_2CuO_{4.12}$ lattice made by $50 \times 50 \times 20$ unit cells. In this model structure, the O-i located with positive correlations along the a- and b-axes has been created by the Monte Carlo method [63] using the software package DISCUS [64].

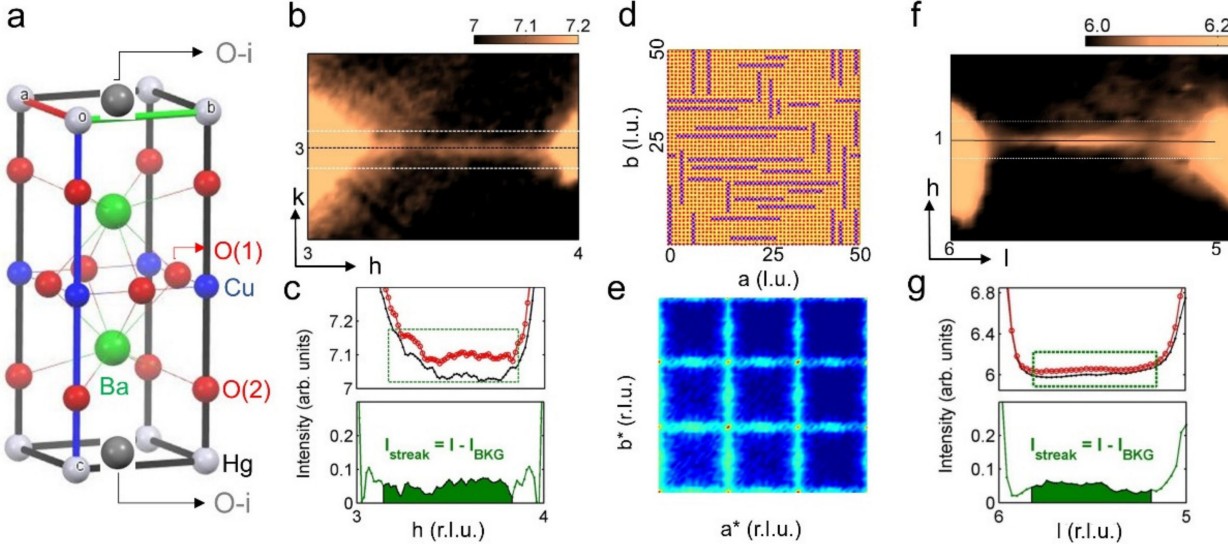

**Figure 1.** (**a**) Schematic representation of the crystal structure of tetragonal $HgBa_2CuO_{4+y}$. The oxygen sites O(1), O(2) and the interstitial O-i are shown. (**b**) A typical streak in the hk plane of the reciprocal space connecting two principal XRD reflections of tetragonal Hg1201 single crystal. (**c**) The diffraction intensity (red circles) along the h direction indicated by the black horizontal line and the background (black points) obtained by merging profiles along (h, $3 \pm \varepsilon$) directions indicated by the white dashed lines in (**b**). The difference between the diffraction intensity and the background, calculated in the h range indicated by the dotted rectangle in (**b**), giving the streak intensity value, $I_{streak}$ (green area) is shown in the lower panel. (**d**) Structural model of ordered O-i in a portion of crystal with $50 \times 50 \times 20$ unit cells. The O-i (violet spheres) are placed in unidimensional chains by the Monte Carlo method considering a positive correlation of 0.5 along the a and b directions with a concentration of 12%. (**e**) Fourier transform of the generated structure for a lattice composed by $50 \times 50 \times 20$ cells in the hk plane of X-ray transmission pattern. We can observe diffuse streaks along the a* and b* directions, in qualitative agreement with diffraction measurements. (**f**) O-i diffuse streak in the hl plane of the reciprocal space connecting the (1, 5) and (1, 6) XRD reflections of Hg1201. (**g**) Diffraction profile (red circles) along the l direction and the background (black points) obtained by merging profiles along ($1 \pm \varepsilon$, l) directions indicated by the white dashed lines in (**f**). The difference between the diffraction profile and the background, calculated in the l range indicated by the dotted rectangle in (**a**), giving the streak intensity value, $I_{streak}$ (green area) is shown in the lower panel.

We obtained qualitative agreement with the experimental diffraction pattern, generating a model of crystal structure with O-i on the $\frac{1}{2}$, $\frac{1}{2}$, and 0 positions, forming stripes along both the a and b directions. Figure 1d shows a pictorial view of O-i arrangement along the a and b crystallographic directions. In Figure 1e, the Fourier transform of the modeled structure is shown and it presents the streaks along the a* and b* directions, in agreement with the measured pattern shown in Figure 1b. This demonstrates that the observed diffuse streaks are determined by the O-i stripes, where the average concentration of O-i is 12% over a lattice of $50 \times 50 \times 20$ unit cells corresponding to about $20 \times 20 \times 20$ nm$^3$.

Scale-free distribution of oxygen wires from microscale to mesoscale has been probed first by scanning micro XRD measurements (SμXRD) and thus scanning nano XRD measurements (SnXRD) to enhance the spatial resolution. Both SμXRD and SnXRD measurements have been carried out at the ID13 beamline of ESRF in Grenoble. The O-i stripes in Hg1201 run along the a(b) direction with no correlation along the c direction; therefore, they give streaks also in the hl plane. The streak intensity, $I_{streak}$, in the hl plane has been calculated for each frame collected in the reflection geometry during the scanning of the sample, similar to what has been carried out for the $I_{streak}$ calculations in the hk plane (see Figure 1f,g).

In this way, we have built the maps of the streak intensity, $I_{streak}$, that visualize the O-i spatial distribution. The maps measured in SμXRD and SnXRD show rich stripe-ordered

O-i (yellow regions) embedded in a matrix of disordered O-i (black regions), as shown in Figure 2a,b, respectively. To characterize this inhomogeneity, we used a spatial statistics approach. More specifically, we have computed the probability density function (PDF) spatial correlations, $G(r)$, and percolation pathways of the streak intensity for both SμXRD and SnXRD. The PDF can be modelled by a power law behavior given by $PDF(I_{streak}) = C(I_{streak})^{-\alpha}$, where C is a constant. We find the critical exponent $\alpha = 2.0 \pm 0.1$ at both micron and mesoscale; the $I_{streak}$ of SnXRD has been scaled in a way that its PDF overlaps the PDF of $I_{streak}$ in the SμXRD, as shown in Figure 2c. The intensity of streak values ranges from values smaller than 1 in the poor oxygen wire regions to values larger than $10^3$ in the rich O-i wire zones. The results show the scale-free organization of O-i in the sample, as it was already found at microscale in other oxygen-doped cuprates. However, we observe that in the SμXRD maps, the size of oxygen wire domains does not exceed the single pixel size corresponding to an area of $1 \times 1$ μm² given by the scanning beam size. This limitation has been overcome by enhancing the real space resolution in XRD setup using SnXRD, where the beam size has been reduced to $0.1 \times 0.3$ μm². Indeed, in this case, the submicron structural features of domains rich in oxygen atomic wires can be appreciated (see Figure 2b).

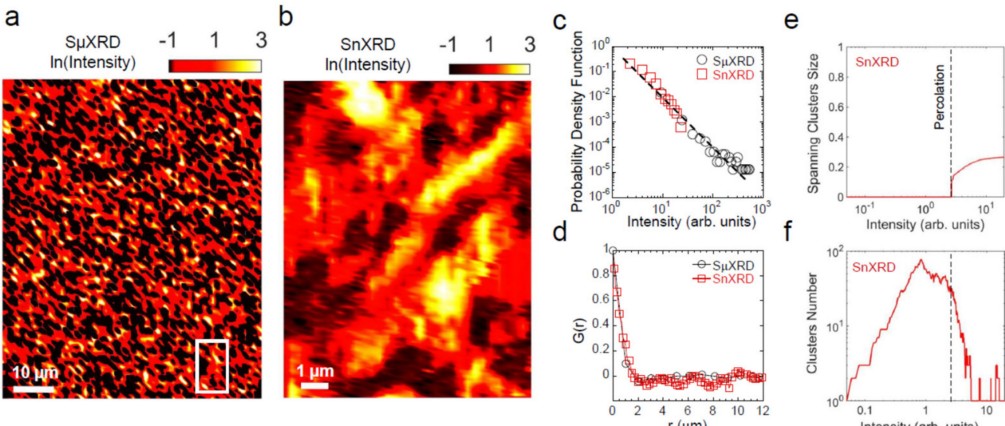

**Figure 2.** (**a**) Logarithmic spatial map of streak intensity extracted by SμXRD with $1 \times 1$ μm² spatial resolution. The rich stripes-ordered O-i (yellow regions) are anticorrelated with CDW rich domains (black regions). The bar corresponds to 10 μm. (**b**) SnXRD spatial map measured with $0.1 \times 0.3$ μm² of streak intensity in logarithmic scale. Here, we can appreciate the interface (red regions) between the rich stripes-ordered O-i (yellow regions) and CDW puddles (black regions). The bar corresponds to 1 μm and the size of the map corresponds to the size of the white rectangle in (**a**). (**c**) Plot of the probability density function of rich stripes-ordered zones relative to the SμXRD (open squares) and SnXRD (full circles) maps in (**a**,**b**). Both distributions have a power law exponent $\alpha = 2$. (**d**) $G(r)$ calculated on the SμXRD (open squares) and SnXRD (full circles) maps in (**a**,**b**). (**e**) Spanning clusters size and (**f**) forming clusters number calculated for the maps in (**b**). The percolation thresholds are represented by the vertical dashed lines.

To quantify the submicron structural features, we have extracted the spatial correlation function $G(r)$ shown in Figure 2d. We observe that the decay of $G(r)$ corresponds to 2 μm in both SμXRD and SnXRD, indicating this distance as the typical size of domains rich in oxygen atomic wires.

We have studied the connectivity between the domains rich in O-i wires using cluster analysis. More specifically, we have calculated all the forming clusters, selecting the cluster with the largest extent, as a function of the streak intensity, $I_{streak}$. When this extent is equal to the system size, there is a spanning cluster and the system percolates. Therefore, we have calculated the percolation threshold, $P(I_{streak})$, the spanning cluster size, and the number of forming clusters in both SμXRD and SnXRD maps in Figure 2a,b. The results are shown in Figure 2e,f. At micron resolution, we cannot find a percolation threshold due to the

resolution limit. On the other hand, in SnXRD map, we can find small clusters forming up to the percolation threshold, P(I$_{streak}$) = 2.7. At this intensity value, a spanning cluster occurs as an intermediate space (red color in the maps of panels a and b) connecting zones rich in oxygen wires (yellow spots in the maps) and CDW puddles (black regions).

Band structure calculations of doped HgBa$_2$CuO$_{4+y}$ at optimum doping with 0.12 < y < 0.16 have been carried out with the assumption of very large 1D and 2D superstructures of ordered O-i [60]. The band structure calculations agree on the strong inhomogeneity of hole doping in metallic CuO$_2$ plane near and far from the O-i oxygen interstitial wire. Following the experimental atomic inhomogeneity of doped HgBa$_2$CuO$_{4+y}$ crystal, its structure is formed by a nematic phase of O-i rich metallic wires made of HgBa$_2$CuO(5) unit cells [CuO(1)$_2$]*[BaO(2)]*[HgO(3)]*[BaO(2)]* separated by undoped portions running in the 100 and 010 directions, made of HgBa$_2$CuO$_4$, tetragonal units [CuO(1)$_2$][BaO(2)][Hg][BaO(2)]. The O-i sites in HgBa$_2$CuO(5) occupy a rather open empty space, which normally is unoccupied in HgBa$_2$Cu$_4$. Possible lattice deformations around O-i impurities are not considered in the calculations [61]. The charge density is strongly inhomogeneous and differs considerably between sites in the proximity and far away from the wire of oxygen interstitials.

Figure 3 shows the local decomposition of the DOS at E$_F$ (in units of (atom eV)$^{-1}$) for Hg$_{12}$Ba$_{24}$Cu$_{12}$O$_{48}$ and "striped" Hg$_{12}$Ba$_{24}$Cu$_{12}$O$_{48+n}$ with n = 2 oxygen interstitials. Apical O2 and planar O1 are the apical Oap and planar Opl oxygen coordinated by Cu. A periodic boundary condition was considered as an artificial periodicity of 6-unit cells. The supercell Hg$_{12}$Ba$_{24}$Cu$_{12}$O$_{48+2}$ extends over 6- and 2- lattice constants along x and y, respectively, and two additional oxygen interstitials were inserted at O(3) site in the Hg-plane forming an atomic stripe running along y. At a distance of 3-unit cells from the oxygen interstitial wire, the local electronic structure becomes close to the undoped system. Panel 3a shows the total density of states (DOS) for superconducting HgBa$_2$CuO$_{4+y}$ with the one-dimensional oxygen interstitials O-i in the [HgO-i] spacer layers and for the undoped cuprate. Figure 3b shows the partial DOS in the apical oxygen sites and Figure 3c the partial DOS in the atomic Hg plane.

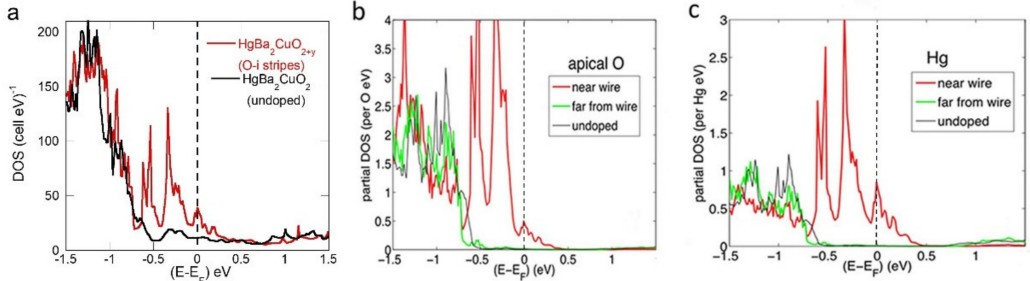

**Figure 3.** (**a**) Total density of states (DOS) for superconducting HgBa$_2$CuO$_{4+y}$ doped by oxygen interstitials O-i, forming one-dimensional oxygen interstitial O-i wires in the HgO$_y$ layers (red curve) showing a DOS peak at the Fermi level (E$_F$) due to a van Hove singularity (VHS). Panels (**b**,**c**) show the partial DOS near and far from the O-i stripes on the apical oxygen sites and on the atomic [Hg] plane (hosting O-i stripes), respectively.

The wire–wire transversal interaction between adjacent wires in the CuO$_2$ plane is weak due to the chosen large distance (6a) along x. The wire–wire interaction along z is weak between the CuO$_2$ planes in this layered cuprate perovskite. Correlation is not expected to be an issue for cuprates with doping well away from half-filling of the d-band in agreement with the angular correlation of positron annihilation radiation (ACAR) spectroscopy [59], which probes FSs and bands that evolve with doping in agreement with density-functional theory (DFT) calculations in the overdoped regime.

The density of states at the Fermi energy, N(E$_F$), is higher than the undoped system, about 3 eV$^{-1}$ per elementary cell. The increase in DOS is limited to the Cu adjacent to the O-i wire; see Figure 3 and Table 1. Ba and Hg atoms are negligible N(E$_F$) in the undoped case, but the striped O-rich system shows a large increase in DOS on the sites near oxygen

interstitials [60]. Hybridization with the p-states on the oxygen interstitials creates a large s- and d-DOS on Hg, and increase in the p-DOS on apical and planar oxygen states. The increase in the local DOS can be understood as an effect due to the O-i atomic stripe, which forms electronic wires with high hole doping when $E_F$ is approaching the wire van Hove DOS peak. The local DOS is very small in more distant atoms in the other atomic layers (BaO, Hg) with values very similar to the undoped Hg1201. At two lattice units from the O-stripe, the DOS already seems to have restored its local character as an undoped Hg1201, except for the modest increase in Cu-d DOS. Moreover, this is corroborated by the electric charge density within the Cu atoms, as well as by an analysis of the folded FS [60].

**Table 1.** Calculated local decomposition of the DOS at $E_F$ within the $CuO_6$ tetrahedra (in units of $(atom\ eV)^{-1}$) for undoped $Hg_{12}Ba_{24}Cu_{12}O_{48}$ and doped $Hg_{12}Ba_{24}Cu_{12}O_{48+n}$ (where n denotes oxygen dopant at O(3) site from ref. [60]. Planar oxygen ions in the O(1) sites of $CuO_2$ plane and apical oxygen ions in the O(2) site are indicated as Opl and Oap, respectively. Total DOS is here for the $CuO_6$ tetrahedron only. The doped oxygen interstitials (O-i) at the O(3) site in the HgO layer form one-dimensional atomic wires. In the doped crystal, the charge density in $CuO(1)_4O(2)_2$ units creates metallic $[CuO_6]^*$ wires that run in the 010 direction. Charge density inhomogeneity is indicated by the peak of the local DOS of 1.76 atom $eV^{-1}$ near the atomic O-i stripes. It appears as if the Fermi level is near a van Hove singularity (VHS) of the local electronic band structure in the overdoped stripes, while the local DOS is 30–35% lower within the separate underdoped portions far from O-i atomic stripes.

| | Cu | $O_{pl}$ | $O_{ap}$ | Total |
|---|---|---|---|---|
| $Hg_{12}Ba_{24}Cu_{12}O_{48}$ | 0.73 | 0.12 | 0.0 | 0.85 |
| $Hg_{12}Ba_{24}Cu_{12}O_{48+2}$ near O-i stripes | 1.0 | 0.21 | 0.50 | 1.71 |
| far from O-i stripes | 0.93 | 0.18 | 0.15 | 1.26 |

## 3. Discussion

We have used scanning nano-X-ray diffraction with a focused beam of $100 \times 300\ nm^2$ for directly imaging the mesoscale texture formed by the oxygen interstitial stripes in $HgBa_2CuO_{4+y}$. The O-i stripes show a scale-free-like organization, as found at microscopic and atomic scales in scanning micro-X-ray diffraction and previous scanning transmission microscopy measurements in different systems. These results demonstrate the self-similarity in the spatial texture of the oxygen interstitial stripes affecting the material properties. This specific ordering of defects changes dramatically the electronic structure with the rise in a sharp peak at the Fermi energy in the density of state. This indicates that the ordering of interstitial oxygen dopants in atomic stripes gives a metallic character to the $CuO_2$ plane. Therefore, the positive effect on $T_C$ from the ordering of O-i might come from the hole doping that the excess O-i provides to the regions far from the dopants. This scenario is consistent with the recent discovery that the space far from the oxygen interstitial atomic stripes in Hg1201 hosts segregated CDW ordered puddles [57]. The compelling evidence of scale invariant distribution of metallic wires in optimum doping Hg1201 supports the proposal that the system showing the superconducting high-critical temperature is in the proximity of a critical point for nanoscale phase separation near a Lifshitz transition in multi-band correlated electronic materials [64–67] in diborides [68], cuprates [69], organics [70,71], pressurized hydrides [72,73] showing high-order van Hove singularity [74,75], which could be further amplified by spin orbit coupling [76] due to internal electric field gradients driven by local charge segregation in oxygen-doped Hg1201. The compelling evidence of the nematic phase with scale-free distribution of atomic wires in mercury-based superconducting perovskites at optimum doping indicate the presence of critical opalescence [77] of atomic wires at a critical point involving local charge, spin [78,79], and electron-phonon interactions in two-dimensional superconductors [11,80]. A quantum critical point connected with charge density waves has been predicted by Castellani and Arpaia [81,82].

## 4. Materials and Methods

High-energy X-ray diffraction. The O-i organization in Hg1201 has been investigated by high-energy X-ray diffraction (XRD) in transmission geometry at the BW5 beamline of the DESY Synchrotron in Hamburg, using $200 \times 200$ µm X-ray beam with 100 KeV from the source by a double-crystal Si(111) monochromator. The high-energy allowed us to probe the average bulk O-i organization. The beam line is equipped with a single-axis diffractometer with a motorized goniometric xyz stage head. The single crystal c-axis has been oriented parallel to the direction of the X-ray incoming beam. In this geometry, we can probe the lattice fluctuations on the ab plane. The diffraction patterns have been collected by an area detector at room temperature. We achieved evidence of diffuse streaks connecting Bragg peaks associated with oxygen interstitial (O-i) atomic stripes.

Scanning micro-X-ray diffraction (SµXRD) experiments were performed in reflection geometry using the ID13 beamline at ESRF, Grenoble, France. We applied an incident X-ray energy of 13 KeV. By moving the sample under a 1-µm focused beam with an x–y translator, we scanned a sample area of $65 \times 80$ µm$^2$, collecting 5200 different diffraction patterns at T = 100 K. For each scanned point of the sample, the intensity profile of the streak from (1,5) to (1,6) Bragg peaks in the hl plane was extracted.

In conclusion, the scanning nano-X-ray diffraction (SnXRD) experimental setup was equipped with a double-crystal monochromator and a Kirkpatrick–Baez mirror system supplied with a beam size of $100 \times 300$ nm$^2$. The data have been recorded using a wavelength of 0.1 nm. A charge-coupled device FreLon area X-ray detector has been used for the experiment. The sample was mounted on a motorized goniometric xyz stage head. In this study, we achieved evidence of diffuse streaks connecting Bragg peaks associated with oxygen interstitial (O-i) atomic stripes. The diffraction patterns have been collected at room temperature and the intensity profile of the streak from (1,5) to (1,6) Bragg peaks was extracted. We scanned a sample area of $9 \times 12$ µm$^2$ moving the sample by means of a piezo yz stage with a step size of 100 and 300 nm along the vertical and horizontal directions, respectively.

**Author Contributions:** Conceptualization, G.C. and A.B.; methodology, A.B.; software, G.C. and M.V.M.; validation, G.C., T.J. and A.B.; resources, G.C. and A.B.; data curation, G.C. and M.V.M.; writing—original draft preparation, A.B., G.C., T.J. and M.V.M.; writing—review and editing, G.C., M.V.M. and A.B.; funding acquisition, A.B. All authors have read and agreed to the published version of the manuscript.

**Funding:** M.V.M. acknowledge the Superstripes-onlus for the RICMASS research fellowship.

**Data Availability Statement:** The data that support the findings of this study are available from the corresponding author, G.C., upon reasonable request.

**Acknowledgments:** We thank the 1D13 beamline staff, Manfred Burghammer, Alessandro Ricci, and Nicola Poccia for the experimental help and we are grateful to D. Zhigadlo, S. M. Kazakov for the crystal synthesis.

**Conflicts of Interest:** The authors declare no conflict of interest. The funders had no role in the design of the study; in the collection, analyses, or interpretation of data; in the writing of the manuscript; or in the decision to publish the results.

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
