# Peer review of "Scale-Free Distribution of Oxygen Interstitial Wires in Optimum-Doped HgBa2CuO4+y"

_condensedmatter, doi:10.3390/condmat7040056_

Round 1

Reviewer 1 Report

This paper shows that the self-similarity presents in the spatial texture of the oxygen interstitials stripes influences the properties of the superconducting oxide. In fact, the ordering of defects changes the electronic structure significantly with the appearance of a DOS sharp peak near the Fermi level. This work reveals important paradigms of doped high temperature superconducting perovskites, thus it deserves to be published. I have noticed small details that must be address before publication. For example,

Line 117: "Istreak", the "steak should be an index.

Line 223: The sentence is incomplete.

Line 237: "The density os states at theFermi energy" should be "The density of states at the Fermi energy".

Line 239: in  "N(EF)", "F" should be a subscript.

To improve the overall formatting, I suggest the authors check subscripts, superscripts, suitable fonts, and other minor stylistic details.

Author Response

I have noticed small details that must be address before publication. For example,

Line 117: "Istreak", the "steak should be an index.

Line 223: The sentence is incomplete.

Line 237: "The density os states at theFermi energy" should be "The density of states at the Fermi energy".

Line 239: in  "N(EF)", "F" should be a subscript.

To improve the overall formatting, I suggest the authors check subscripts, superscripts, suitable fonts, and other minor stylistic details.

Reply. We thank the referee for his positive comments and for noticing the details above. The typos have been corrected in the lines 117 223 237 239.

Reviewer 2 Report

In this paper, the authors report on an experimental investigation of the nonuniform spatial distribution of interstitial oxygen atoms, O-i, in superconducting Hg-1201 single crystals by means of scanning micro- and nano-x-ray diffraction measurements supported by ab initio calculations of the electronic structure.

The main result of the paper is the observation of diffuse scattering features suggesting the formation of a filamentary-like arrangement of the O-i’s along the a- and b- unit cell axis in the mesoscale. The authors’ data analysis indicates that the spatial correlation function governing this medium-range ordering obeys a scale-free distribution with power-law exponent -2. The authors therefore propose a percolation scenario of superconducting currents flowing along the filaments. The ab initio calculations on supercells that simulate the ordering of the interstitial oxygens support the above scenario, for they predict a significant spatial modulation of the density of states moving away from the filaments.

My overall judgement is that the paper contains interesting results that convincingly show the crucial role of chemical and electronic inhomogeneities in cuprates on the normal and superconducting properties. The choice of the Hg-1201 as model system is well justified by the authors, considering the comparatively simple crystal structure. The experimental data are of good quality and the data analysis convincing. Having said this, I should suggest a number of improvements. Main points are as follows.

(i) In the introduction, the notion of scale-free or power-law distribution in granular or disordered matter should be first explained for the sake of clarity.

(ii) In the introduction, the notion of scale invariance near a quantum critical point and its possible connection with unconventional superconductivity deserves to be further developed, for this is an open and interesting question.

(iii) In the discussion section, the authors use the alternative notion of “multiscale invariance” after having used the notion of “scale invariance”. Please explain the difference, if any.

(iv) For the readability of the paper, I would suggest reducing the number of cited references. For example, by referring to optimally doped Hg-1201 (line #53), I do not think as many as thirteen references are necessary.

(v) I counted as many as 36 self-citations out of 80 total references. I strongly recommend reducing this number to about 10% in order to strengthen the credibility of a good paper.

(vi) The manuscript requires to be written more accurately. I should recommend a thorough revision to remove numerous imperfections including:

·       an incomplete sentence (line #223)

·       missing units in Table I

·       numerous superscripts and subscripts are not properly typed

·       define EF (line #204)

·       the author contribution section is incomplete

·       a few expressions are not appropriate or probably wrong, for example in the sentence “... has been attracting interest now days...” (line #35). Therefore, a revision of the text by a native English speaker may be useful.

(vii) Line #57: the expression 'vertical direction' is ambiguous. Do the authors mean the b-axis (010) or the c-axis (001) direction?

In conclusion, I recommend publication of the paper provided the authors make the above recommended changes.

Author Response

(i) In the introduction, the notion of scale-free or power-law distribution in granular or disordered matter should be first explained for the sake of clarity. (ii) In the introduction, the notion of scale invariance near a quantum critical point and its possible connection with unconventional superconductivity deserves to be further developed, for this is an open and interesting question.

Reply. We have replaced two references, 23, 24 where the scale-free, power-law distributions and scale invariance in granular or disordered matter are explained.

(iii) In the discussion section, the authors use the alternative notion of “multiscale invariance” after having used the notion of “scale invariance”. Please explain the difference, if any.

Reply. We have replaced the wording “multiscale invariance” with “scale free distribution” of oxygen interstitial wires.

(iv) For the readability of the paper, I would suggest reducing the number of cited references. For example, by referring to optimally doped Hg-1201 (line #53), I do not think as many as thirteen references are necessary.

Reply. We think that is important for the reader to remind relevant papers on the Hg1201 topic made over the last 3 decades of research in this field.

(v) I counted as many as 36 self-citations out of 80 total references. I strongly recommend reducing this number to about 10% in order to strengthen the credibility of a good paper.

Reply. We have removed some self-citations; however, the others are needed for the reader to understand the results presented in this work.

(vi) The manuscript requires to be written more accurately. I should recommend a thorough revision to remove numerous imperfections including:

  • an incomplete sentence (line #223)
  • missing units in Table I
  • numerous superscripts and subscripts are not properly typed
  • define EF (line #204)
  • the author contribution section is incomplete
  • a few expressions are not appropriate or probably wrong, for example in the sentence “... has been attracting interest now days...” (line #35). Therefore, a revision of the text by a native English speaker may be useful.

Reply. We have revised the manuscript; many typos and mistakes have been corrected.

(vii) Line #57: the expression 'vertical direction' is ambiguous. Do the authors mean the b-axis (010) or the c-axis (001) direction?

Reply.We have replaced the expression 'vertical direction' with “b-direction (010) of the ab plane”

Reviewer 3 Report

In this manuscript, Campi et al. present a scanning X-ray diffraction investigation of HgBa2CuO4+y (Hg1201) single crystals, both at the microscale (SμXRD) and at the nanoscale (SnXRD). Object of the investigation is the texture of oxygen interstitial (O-i) atoms within the system, to be determined down to the nanoscale. The relevance is given by the spatial anticorrelation existing between these regions and the patches of charge density waves (CDW), proven by different previous works. Being so intertwined, O-I can affect the structure of CDW, which have been recently associated to the transport properties of cuprate high-Tc superconductors (HTS) both in the normal and in the superconducting states; and together, O-I and CDW are the natural candidates to be the sources of the electron nematicity, which is characterizing the ground state of HTS already at room temperature. Taking advantage of several statistics tool for the data analysis, the authors i) confirm at the microscale the striped configuration of O-i, already shown in the past in other HTS compounds; ii) visualize at the nanoscale a percolation pattern between CDW patches and O-i stripes. The manuscript is interesting and it will be of interest to the community. However, in order to bring their manuscript to the level required for publication on “Condens. Matter”, the authors need to work on same aspects that I feel will significantly improve the quality of the paper:

1. The English is sloppy in several sections, and the manuscript in general requires a strong polishing on many aspects, from the figures (whose poor quality makes them often not easy to read) to the formatting (subscripts and superscripts are almost absent throughout the whole manuscript), to the text itself (a sentence is incomplete, see line 224, and many evident misspells are still present, even in the acknowledgement section – 1D13). The authors are strongly encouraged to very carefully proofread the manuscript, since the general feeling is that is hasn’t been done prior to the first submission, improve the style in particular in the introduction section, and fix the general formatting of the manuscript before the next resubmission.

2.   Even though, as previously stated, the accomplished research is relevant, for the general reader it would be very tough to understand from the introduction what should be gained from the knowledge of the spatial texture of the O-i at the nanoscale. First of all, the presence of anisotropic structures at room temperature, and the understanding of their properties, is crucial to explain the still elusive room temperature nematicity observed in HTS (Nature 547, 432 (2017); Science 373, 1506 (2021)). No mention is given about this implication. Moreover, the authors should take into consideration that – while the puddles on the um scale are a universal property of HTS – the nm-scale details might in principle depend on the sample homogeneity and quality of the samples: this is an issue for HTS materials, for their well-known oxygen out diffusion and sensitivity to defects and disorder, which tends to be enhanced in vacuum and under X-ray beam. How can we exclude the observed details of O-i at the nanoscale are intrinsic properties of Hg1201?

3. The beam used for SnXRD is 100x300 nm2. Are we losing any info below 100 nm which might modify the percolation picture described by the authors?

4.  In the beginning of section 2, a picture of the Hg1201 unit cell, highlighting the real space positions of the O-i atoms, would help the understanding of the object of the authors investigation.

5. The authors find, both at the um and at the nm scale, an exponent alpha=2 ruling the power law behavior of the density function. In the text, it is not clear if and how the absolute value of this exponent is connected to the spatial distribution of the O-i chains.

6.  The observed critical opalescence of the O-i stripes has been interpreted by the authors in terms of a proximity to a quantum critical point (QCP). However, here the conclusions from the authors i) does look not properly contextualized ii) neither strongly supported by the data. Regarding the point i), the scenario suggested by the authors is very appealing, since in the HTS at the optimally doped regime a QCP connected to the CDW has been theoretically predicted (Phys. Rev. Lett. 75, 4650 (1995)) and experimentally measured (Science 365, 906 (2019)). And the strong (anti)correlation between CDW and O-i atoms might suggest an active role of O-i atoms in the determination of this QCP: this could be a strong point toward their interpretation, but no paper about the CDW QCP is mentioned, which might support their scenario. Regarding point ii), I suggest the authors to slightly weaken their statement, since their data are only taken at the optimally doped level. A connection between their percolation scenario and the QCP could be strongly supported only performing their measurements at other doping levels, and proving that this occurrence is only present close to the optimally doped regime.

Author Response

  1. The English is sloppy in several sections, and the manuscript in general requires a strong polishing on many aspects, from the figures (whose poor quality makes them often not easy to read) to the formatting (subscripts and superscripts are almost absent throughout the whole manuscript), to the text itself (a sentence is incomplete, see line 224, and many evident misspells are still present, even in the acknowledgement section – 1D13). The authors are strongly encouraged to very carefully proofread the manuscript, since the general feeling is that is hasn’t been done prior to the first submission, improve the style in particular in the introduction section, and fix the general formatting of the manuscript before the next resubmission.

Reply: We have revised the manuscript; many typos have been corrected.

  1. Even though, as previously stated, the accomplished research is relevant, for the general reader it would be very tough to understand from the introduction what should be gained from the knowledge of the spatial texture of the O-i at the nanoscale. First of all, the presence of anisotropic structures at room temperature, and the understanding of their properties, is crucial to explain the still elusive room temperature nematicity observed in HTS (Nature 547, 432 (2017); Science 373, 1506 (2021)). No mention is given about this implication. Moreover, the authors should take into consideration that – while the puddles on the um scale are a universal property of HTS – the nm-scale details might in principle depend on the sample homogeneity and quality of the samples: this is an issue for HTS materials, for their well-known oxygen out diffusion and sensitivity to defects and disorder, which tends to be enhanced in vacuum and under X-ray beam. How can we exclude the observed details of O-i at the nanoscale are intrinsic properties of Hg1201?

Reply We have considered this point in the introduction and we have inserted the two references, 61, 62

  1. The beam used for SnXRD is 100x300 nm2. Are we losing any info below 100 nm which might modify the percolation picture described by the authors?

Reply In our observations we have improved the spatial resolution in O-i arrangement visualization using a beam size of 100x300 nm2. This is the minimum resolution needed to observe percolation of O-I wires.

  1. In the beginning of section 2, a picture of the Hg1201 unit cell, highlighting the real space positions of the O-i atoms, would help the understanding of the object of the authors investigation.

Reply We have added the panel (a) in Figure 1 to show the crystal structure of Hg1201, highlighting the real space positions of the O-i atoms.

  1. The authors find, both at the um and at the nm scale, an exponent alpha=2 ruling the power law behaviour of the density function. In the text, it is not clear if and how the absolute value of this exponent is connected to the spatial distribution of the O-i chains.

Reply. The absolute value of the exponent alpha fully characterize the power law distribution of the intensity of the ordered O-I, as it is explained in the new added reference [23].

  1. The observed critical opalescence of the O-i stripes has been interpreted by the authors in terms of a proximity to a quantum critical point (QCP). However, here the conclusions from the authors i) does look not properly contextualized ii) neither strongly supported by the data. Regarding the point i), the scenario suggested by the authors is very appealing, since in the HTS at the optimally doped regime a QCP connected to the CDW has been theoretically predicted (Phys. Rev. Lett. 75, 4650 (1995)) and experimentally measured (Science 365, 906 (2019)). And the strong (anti)correlation between CDW and O-i atoms might suggest an active role of O-i atoms in the determination of this QCP: this could be a strong point toward their interpretation, but no paper about the CDW QCP is mentioned, which might support their scenario. Regarding point ii), I suggest the authors to slightly weaken their statement, since their data are only taken at the optimally doped level. A connection between their percolation scenario and the QCP could be strongly supported only performing their measurements at other doping levels, and proving that this occurrence is only present close to the optimally doped regime.

Reply. We have cited the important paper on critical opalescence, ref 77, to make clear the concept to the reader. The quantum critical point connected with charge density waves has been predicted as described in the inserted refs. 80, 81.